# PARP Inhibition Increases the Response to Chemotherapy in Uveal Melanoma

**DOI:** 10.3390/cancers11060751

**Published:** 2019-05-29

**Authors:** Leanne de Koning, Didier Decaudin, Rania El Botty, André Nicolas, Guillaume Carita, Mathieu Schuller, Bérengère Ouine, Aurélie Cartier, Adnan Naguez, Justine Fleury, Vesselina Cooke, Andrew Wylie, Paul Smith, Elisabetta Marangoni, David Gentien, Didier Meseure, Pascale Mariani, Nathalie Cassoux, Sophie Piperno-Neumann, Sergio Roman-Roman, Fariba Némati

**Affiliations:** 1RPPA Platform, Department of Translational Research, Institut Curie, PSL University, 75248 Paris, France; berengere.ouine@curie.fr (B.O.); aurelie.cartier10@laposte.net (A.C.); 2Laboratory of Preclinical Investigation, Department of Translational Research, Institut Curie, PSL University, 75248 Paris, France; rania.el-botty@curie.fr (R.E.B.); guillaume.carita@gmail.com (G.C.); mathieu.schuller@curie.fr (M.S.); adnan.naguez@curie.fr (A.N.); justine.fleury04@gmail.com (J.F.); elisabetta.marangoni@curie.fr (E.M.); fariba.nemati@curie.fr (F.N.); 3Department of Medical Oncology, Institut Curie, 75248 Paris, France; sophie.piperno-neumann@curie.fr; 4Department of Tumor Biology, Institut Curie, 75248 Paris, France; andre.nicolas@curie.fr (A.N.); didier.meseure@curie.fr (D.M.); 5Novartis Institutes for BioMedical Research, Translational Clinical Oncology, Cambridge, MA 02139, USA; vesselina.cooke@novartis.com (V.C.); andrew.wylie@novartis.com (A.W.); 6Oncology IMED, AstraZeneca, 1 Francis Crick Avenue, Cambridge Biomedical Campus, Cambridge CB2 0AA, UK; paul.smith@astra-zeneca.com; 7Genomics Platform, Department of Translational Research, Institut Curie, PSL University, 75248 Paris, France; david.gentien@curie.fr; 8Department of Surgery, Institut Curie, 75248 Paris, France; pascale.mariani@curie.fr; 9Department of Ophthalmologic Surgery, Institut Curie, University of Medicine René Descartes Paris V, 75248 Paris, France; nathalie.cassoux@curie.fr; 10Department of Translational Research, Institut Curie, PSL University, 75248 Paris, France; sergio.roman-roman@curie.fr

**Keywords:** PARP inhibitor, treatment combinations, uveal melanomas, patient-derived xenografts

## Abstract

Uveal melanoma (UM) remains without effective therapy at the metastatic stage, which is associated with *BAP-1* (BRCA1 associated protein) mutations. However, no data on DNA repair capacities in UM are available. Here, we use UM patient-derived xenografts (PDXs) to study the therapeutic activity of the PARP inhibitor olaparib, alone or in combination. First, we show that the expression and the activity of PARP proteins is similar between the PDXs and the corresponding patient’s tumors. In vivo experiments in the PDX models showed that olaparib was not efficient alone, but significantly increased the efficacy of dacarbazine. Finally, using reverse phase protein arrays and immunohistochemistry, we identified proteins involved in DNA repair and apoptosis as potential biomarkers predicting response to the combination of olaparib and dacarbazine. We also observed a high increase of phosphorylated YAP and TAZ proteins after dacarbazine + olaparib treatment. Our results suggest that PARP inhibition in combination with the alkylating agent dacarbazine could be of clinical interest for UM treatment. We also observe an interesting effect of dacarbazine on the Hippo pathway, confirming the importance of this pathway in UM.

## 1. Introduction

Uveal melanoma (UM) is a rare tumor affecting 7/1.0 million of the Western population per year [1] for which no effective systemic therapies exist at the metastatic stage. Dacarbazine (DTIC), which is an alkylating agent leading to DNA lesions, is often the standard arm in trials in metastatic UM, but response rate remains disappointing. Several studies have thus focused on the molecular characterization of UM, with the aim to identify new subpopulations or therapeutic targets [2,3,4,5,6,7]. Well-described recurrent mutations have been observed, such as the *GNAQ* or *GNA11* genes, which are mutated in about 80% of UM and lead to a constitutive activation of the MAPK (mitogen activated protein kinase) and Hippo pathways [8,9]. Nevertheless, the MEK1/2 inhibitor selumetinib, in combination with dacarbazine, fails to improve progression-free survival of metastatic UM [10]. Various targeted therapies have therefore been tested in combination with selumetinib [11] or with the PKC (protein kinase C) inhibitor AEB071 [12] in UM preclinical models, and particularly in patient-derived xenografts (PDXs). On this basis, this last compound is currently being tested in a clinical trial, currently in combination with a MDM2 inhibitor (NCT02601378).

Another recurrent mutation in UM is the *bap1* (*BRCA1-associated protein 1*) gene mutation [13], which is associated with a higher risk of metastases [14,15]. Sporadic germline *bap1* mutations confer a predisposition to several types of cancer, including UM [16], confirming its role in tumorigenesis. BAP1 is a deubiquitinating enzyme involved in chromatin structure, cell cycle progression, and differentiation. Although the direct interaction of BAP1 with BRCA1 remains controversial [17], several studies report a potential role of BAP1 in DNA damage repair, and particularly in homologous recombination (HR). BAP1 is recruited to double strand breaks (DSB) and promotes DNA repair and survival after DNA damage induction [18,19,20]. BAP1 recruitment would be poly(ADP-ribose) polymerase (PARP) dependent [18] and BAP1 function would be required for efficient recruitment of BRCA1 and RAD51 to DNA repair foci [19,20]. In conclusion, BAP1 deficiency might lead to impaired DSB repair by HR and could potentially increase the reliance on parallel repair pathways in a similar manner as BRCA1 deficiency.

Given the potential role of BAP-1 in DNA repair and the frequent administration of dacarbazine, it is surprising that the DNA repair pathways and their therapeutic potential have not yet been evaluated in UM. We hypothesized that PARP may be an interesting target in UM, alone or in combination with other therapies. Indeed, the PARP proteins catalyze the transfer of ADP-ribose to target proteins. PARPs play an important role in various cellular processes and notably in DNA repair by base-excision repair (BER) and nucleotide excision repair (NER). BER and NER are required for repair of DNA lesions induced by certain chemotherapeutic agents and PARP inhibition is therefore an attractive therapeutic option in combination with chemotherapy [21]. Cells displaying deficiencies in HR particularly rely on PARP and are thus remarkably sensitive to PARP inhibition [22]. Whether this could be the case in BAP-1 mutated melanoma has not been assessed. PARP expression and activity has not been extensively studied in UM. One study showed varying mRNA and protein expression levels of PARP1, as well as PARP1 enzymatic activity in five UM cell lines [23]. Similarly, in a small series of 12 UM, a slight and variable perivascular PAR staining has been observed [24]. Hence, it remains to be demonstrated whether PARP could be a therapeutic target in UM patients.

Here, we explore PARP expression in both UM patient’s tumors and a unique panel of PDXs, and we evaluate for the first time the therapeutic potential of the PARP inhibitor olaparib used alone or in various combinations in UM PDXs. Next, using RPPA (reverse phase protein array), WB (western blots), and IHC (immunohistochemistry) analyses, we explored predictive factors that are implicated in the additive effect of olaparib + dacarbazine, as well as protein modifications observed in treated tumors. Hence, our study may be a pivotal preclinical study for the clinical application of PARP inhibitor in the treatment of metastatic UM patients.

## 2. Results

### 2.1. Basal Gene and Protein Expression of PARPs in Patient’s Tumors and Corresponding PDXs

The expression of *PARP* family genes was evaluated using data generated from previously reported Affymetrix GeneChip-Human Exon 1.0 ST arrays, including 12 patient tumors and 32 PDXs (15 at passage one, 13 at passage four, and four at passage nine). Two control genes were used to assess positive and negative expression, i.e., the *MAGE1* gene (negative expression) and the *MGAT5* gene (positive expression). We observed variable expression levels among all *PARP* family genes, with high expression of *PARP1*, *PARP4*, *PARP6*, *PARP10*, *PARP12*, and *PARP14* genes. Moreover, we did not observe variations in gene expression levels between patient’s tumors and their corresponding PDXs at three different in vivo passages (Figure 1 and Appendix A). 

Similarly, we studied protein expression of the p116 uncleaved PARP, the two p25 and p89 cleavage products of PARPs, and the two corresponding ratios cleaved/uncleaved PARPs, using RPPA. Patient’s tumors (P0) and corresponding PDXs at various in vivo passages (P1, P4, and P9) were studied (ten pairs P0–P1 or P0–P4, four pairs P0–P9, 13 pairs P1–P4, and six pairs P4–P9). All results are presented in Figure 1 and in Appendix A. We conclude that PARP protein expression and cleavage was stable between patient’s tumors and their corresponding PDXs at early in vivo passages (P1 and P4). For passage nine, only four PDXs have been studied, and results are therefore less robust, but even at this late passage little variation is visible.

### 2.2. Olaparib Increased DTIC Antitumor Activity in UM PDXs

Various olaparib-based combinations with chemotherapies or targeted therapies were tested in 11 unique well-characterized UM PDXs (Appendix A) that were already used for pharmacological assessments [11,12,25,26,27] and that well reproduce patients’ tumors [28,29].

First, we performed a global comparison of all tested treatments in the various sets of in vivo experiments. Indeed, olaparib was tested in ten models (137 mice); DTIC and fotemustine, which are the two main chemotherapies used in metastatic UM patients, were tested in six models (47 mice) and two models (17 mice), respectively; everolimus, AEB071, and CGM097, that have already showed preclinical efficiency in UM PDXs [12,26], were tested in one (eight mice), three (23 mice), and five (12 mice) models, respectively; finally, AZD0156 and AZD6738, both treatments that may impact DNA repair, were tested in five PDX models (14 and 16 mice, respectively). In all these experiments, as shown in Figure 2, an overall response rate (ORR) below −0.5 and −0.9 was observed in 16% and 0% of mice after olaparib; 43% and 15% after DTIC; 59% and 41% after fotemustine; 75% and 0% after everolimus; 43% and 4% after AEB071; 33% and 0% after CGM097; 20% and 0% after AZD 156; and 33% and 0% after AZD6738. In conclusion, in all tested UM PDXs, olaparib monotherapy was significantly less efficient than DTIC (*p* < 10^−3^), fotemustine (*p* < 10^−3^), everolimus (*p* < 10^−3^), and AEB071 (*p* < 10^−2^), and no significant differences were observed between olaparib and CGM097, AZD0156, and AZD6738.

We hypothesized that olaparib could have an antitumor effect when combined with other therapies and notably with DNA damaging agents, as has been described in other tumor types [30]. We therefore performed a comparison of the ORR of each olaparib-based combination (i.e., olaparib + DTIC, fotemustine, everolimus, AEB071, CGM097, AZD0156, or AZD6738) versus the corresponding compound alone (Figure 2). Only the combination of olaparib + DTIC showed a significantly improved response rate compared to DTIC monotherapy, both for an ORR lower than −0.5 and −0.9 (*p* < 10^−2^ and 0.05, respectively). The ORR ≤ −0.5 increased from 43% for monotherapy to 68% in the combination therapy. Similarly, the ORR ≤ −0.9 increased from 15% to 32%. Six models (MP41, MP55, MP77, MM33, MM52, and MM66) were treated with the olaparib + DTIC combination (Figure 3). For these six experiments, the median RTV of all treated mice was 10.1, 7.6, 5.6, and 1.9 for the control, olaparib, DTIC, and olaparib + DTIC groups, respectively. Similarly, the median probability of progression (doubling time) was 8, 10, 11.5, and 31 days for the control, olaparib, DTIC, and olaparib + DTIC groups, respectively. In conclusion, olaparib significantly increases the antitumor activity of DTIC in UM PDXs, as shown by an increased response rate, reduced tumor volume and prolonged doubling time. In contrast, no significant benefit was observed in other tested olaparib-based combinations (Appendix A).

### 2.3. RPPA-Based Proteomic Study of Olaparib + DTIC Combination in Four UM PDXs

The aim of the proteomic RPPA-based study was to evaluate the correlation of treatment efficacy (olaparib alone, DTIC alone, and olaparib + DTIC) with the expression and activation of various proteins, in order to identify proteins that may play a role in the additive effect of olaparib combined with DTIC. We selected proteins involved in DNA repair, apoptosis induction, and MAPK, Pi3K- and Hippo-signaling pathways. Among the six PDXs that received the olaparib + DTIC combination, four of them were included in the RPPA study, i.e., MP55, MP77, MM33, and MM52. Four to five tumors collected at the end of in vivo experiments and under therapies were studied in each group (control, olaparib, DTIC, and olaparib + DTIC). Overall, the combination of olaparib + DTIC showed an additive effect in three models (MP55, MP77, MM33), while the fourth did not (MM52), with a total number of 15 and seven tumors in the two respective groups.

First, we aimed to define predictive protein markers for the additive efficacy of DTIC + olaparib. For this, we defined proteins whose expression are predictive for response to DTIC alone (MP55, MP77, and MM52), olaparib alone (MM33), and DTIC + olaparib (MP55, MP77, and MM33) in the untreated (control) mice. We then determined proteins that are exclusively correlated to the additive effect of the combination and not to each monotherapy (Figure 4, Appendix A). Interestingly, the main involved proteins belong to the DNA-repair function, with particularly high expression of both PARP and c-PARP proteins, as well as p-FANCD2, p-FANCD2/FANCD2, P53, p-P53, and low expression of ATM, p-ATM, and p-Topo.IIa/Topo.IIa proteins. We also underline the predictive value of low Bcl-X_L_ protein (*p* = 5.1×10^−4^) (Figure 4). 

We evaluated protein expression modifications under the different therapies by comparing untreated mice with treated mice for each PDX model. We observed modifications in DNA repair proteins, both in homologous recombination (HR) and nonhomologous end joining (NHEJ) pathways (Appendix A), as illustrated in Figure 5 for one model showing a synergy between DTIC and olaparib (MP55) and one model showing no synergy (MM52). We did not identify modifications that could be correlated to the additive effect of DTIC + olaparib compared to DTIC alone. 

When looking at apoptosis induction studied by PARP and cleaved-PARP expressions, we did not observe significant variations between tumors collected after DTIC or DTIC + olaparib treatments, except in the MM33 PDX (Appendix A and Appendix A). Moreover, we did not observe significant variations of cleaved-PARP expression after DTIC administration, except in the MM52 PDX. Finally, the study of RPPA-based apoptosis-related protein expression (Bcl-2, Bcl-X_L_, Mcl1, Bax, and Bak) showed that both anti- and proapoptotic protein expressions were modified after DTIC + olaparib administration, in comparison to the control group (Appendix A and Appendix A). These data were confirmed by Western Blot (Appendix A; Appendix A) and IHC (Appendix A) and suggest that apoptosis may not be implicated in the additive effect observed after DTIC + olaparib treatment.

In addition, both MAPK and Pi3K signaling pathways were studied by RPPA, showing no significant recurrent protein expression variations between DTIC- and DTIC + olaparib-treated tumors (Appendix A; Appendix A). This was confirmed by western blot (Appendix A) and IHC (Appendix A).

Finally, we have also studied Hippo-related protein modifications in the four selected PDXs (Appendix A and Appendix A). We observed a highly significant increase of p-YAP (phospho-Yes-associated protein) expression in most mice upon DTIC treatment (Appendix A). These observations were confirmed by IHC, where we observed a high and significant increase of both nuclear and/or cytoplasm YAP and TAZ expression under olaparib alone, DTIC alone, and olaparib + DTIC in all but one (MM33) PDXs (Figure 6 and Appendix A). This result suggests that DTIC does not only induce DNA damage but might also exert an antitumor effect through YAP phosphorylation. 

## 3. Discussion

In this work, we aimed to evaluate the therapeutic potential of the PARP inhibitor olaparib in UM PDXs for the first time. Because PARP expression or activity is potentially predictive for response to PARP inhibition [31], we first confirmed that the PARP gene family is expressed and activated at similar levels in our panel of 11 well-characterized PDX models compared to corresponding patient samples. This is the case at least for the PARP 1–4 genes, which are targeted by olaparib [32]. Next, we showed that olaparib alone was not efficient, but significantly increased the response to the alkylating agent dacarbazine. In contrast, such an additive effect was not observed when olaparib was combined with other compounds, i.e., the alkylating agent fotemustine, the mTORC1 inhibitor everolimus, the PKC inhibitor AEB071, the MDM2 inhibitor CGM097, the ATM inhibitor AZD0156, and the ATR inhibitor AZD6738. 

An additive activity of PARP inhibitor combined with chemotherapeutic agents has already been reported, even in the absence of BRCA1/2 mutation, for the isoquinoline alkylating agent trabectedin [33], camptothecins [34], and, of course, temozolomide/dacarbazine [34,35,36]. The increased in vivo activity of dacarbazine in UM PDXs when combined with olaparib could be explained by the fact that PARP inhibition increases the antitumor effects of DNA-damaging through disruption of the base excision repair pathway function, as previously reported [37]. In addition, BAP1 deficiency might confer a moderate DNA repair deficiency, in a similar manner as BRCA1 deficient tumors, and thus confer an increased sensitivity to olaparib upon DNA damage induction by DTIC. However, among our three models that show an additive effect between olaparib and DTIC, only one (MP55) had a detectable BAP1 mutation (Appendix A). Although the number of models is too low to draw conclusions, olaparib might affect tumor growth independently of BAP1 status, as has been observed in malignant pleural mesothelioma [38]. Alternatively, BAP1 deficiency may not only be induced by mutations. Several studies have indeed reported the absence of nuclear BAP1 protein despite the presence of a wild-type gene [39,40,41,42] and normal mRNA levels [43], suggesting that post-translational regulation, for example by miR31 [44,45], can also induce functional BAP1 deficiency. However, this type of regulation does not seem to be at play in our models, since BAP1 mutation status and IHC staining are concordant. Surprisingly, we did not observe a similar synergy between olaparib and fotemustine. This may be explained by a slightly different mode of action between the two compounds, or by the fact that fotemustine + olaparib was tested in only two PDXs (Appendix A) and statistical power may thus be insufficient to detect an additive effect. 

We also aimed to identify biomarkers of prediction for an additive efficiency of dacarbazine combined with olaparib. While response to DTIC alone seems to be associated with decreased activation of PI3K/Akt and MEK/ERK signaling and decreased expression of components of the Hippo pathway (MST1/2 and LATS), we have identified two classes of proteins predictive for the combination of dacarbazine with olaparib: those involved in DNA repair (high expression of PARP, c-PARP, p-FANCD2, p-FANCD2/FANCD2, P53, and p-P53, and low expression of ATM, p-ATM, and p-Topo.IIa/Topo.IIa), and those involved in apoptosis (low Bcl-X_L_ protein). High PARP expression has already been identified as being predictive for the synergy between olaparib and the alkylating agent trabectedin [33]. More generally, several proteins involved in DNA damage repair are known as predictive biomarkers for PARP inhibition [46,47]. We also showed that a low expression of the antiapoptotic Bcl-X_L_ protein was a highly predictive marker for the additive efficacy of dacarbazine + olaparib (*p* = 5.1 × 10^−4^). Our observation is concordant with that of Engert and colleagues, showing that olaparib + temozolomide combination induced downregulation of the antiapoptotic protein MCL-1, and the activation of the two proapoptotic proteins BAX and BAK with apoptosis occurrence (mitochondrial outer membrane permeabilization and caspase-3 activation) [35], altogether suggesting the possible role of apoptosis-regulatory proteins as predictive markers of response or proteins that are impacted by such treatments. This observation suggests that a Bcl-X_L_ inhibitor may be useful to sensitize tumors to the concomitant administration of dacarbazine + olaparib. 

Finally, we have also made a very striking observation in the view of UM. It was previously reported that *GNAQ*/*GNA11* mutations induce dephosphorylation and thus nuclear accumulation of the YAP protein [19,48], both through a Hippo-independent and a Hippo-dependent pathway [8,9]. Verteporfin, known to disrupt the interaction between YAP and TEAD4, was indeed able to induce apoptosis of UM cells [49]. In our study, we observed a strong and significant increase of both YAP and TAZ protein expression and phosphorylation upon treatment with DTIC and, to a lesser extent, olaparib. Those results therefore suggest that these drugs may exert their antitumor effect also through phosphorylation and thus inactivation of YAP. Administration of verteporfin in addition to DTIC + olaparib might be a path to explore in order to further potentiate the synergy of the combination treatment.

## 4. Materials and Methods

### 4.1. Uveal Melanoma Preclinical Models

Eleven PDXs representative of the UM disease were used, five obtained from primary intraocular tumors (MP34, MP41, MP42, MP55, and MP77), five from liver metastatic tumor samples (MM26, MM52, MM66, MM224, and MM252), and one from cutaneous metastasis (MM33). Molecular features of these models are presented in the Appendix A. Some of these PDXs have already been reported [28,29].

### 4.2. Compounds

The PARP inhibitor olaparib, partially provided from Novartis Institutes for Biomedical Research (NIBR, Cambridge, UK) and Astra-Zeneca (Oncology Bioscience, Cambridge, UK), was orally administered at a daily dose of 50 mg/kg or 100 mg/kg, five days a week, depending to the tolerance of tested combinations. Two cytotoxic agents were tested, both alkylating agents, i.e., dacarbazine (Deticene^®^ or DTIC) (Medac, Lyon, France) and fotemustine (Muphoran^®^) (IRIS, Orly, France). Five different targeted therapies were used, the mTORC1 inhibitor everolimus (RAD001, Certican^®^) (Novartis, Basel, Switzerland), the pan-PKC inhibitor AEB071 (provided from Novartis), the MDM2 inhibitor CGM097 (provided from Novartis), the ATM inhibitor AZD0156 (provided from Astra-Zeneca), and the ATR inhibitor AZD6738 (provided from Astra-Zeneca). All treatment schedules are presented in the Appendix A. 

### 4.3. In vivo Tumor Growth and Antitumor Efficacy

For in vivo therapeutic studies, female SCID mice (Janvier Labs, Le Genest Saint Isle, France) were xenografted with a tumor fragment of 20–40 mm^3^. Mice bearing growing tumors with a volume of 60–150 mm^3^ were randomly assigned to the control or treatment groups. The number of animals per group was related to the date of experiments. While initial in vivo experiments were performed with about seven to ten mice per group, final experiments, for ethical reasons and to decrease the cost of the study, were performed according to a related- “single-mouse” schedule in which three mice bearing growing tumor were included per group [11,50]. Overall, due to the fact that this study was conducted during a relatively long period, about two years of in vivo experiments, the number of models used for each combination of treatments and the number of mice per model was not homogeneous along the study. Treatments were started on day one. Mice were weighed and tumors measured twice a week. Xenografted mice were sacrificed when their tumor reached a volume of 2500 mm^3^. 

Mice bearing tumors with a volume from 50 to 150 mm^3^ were individually identified and randomly assigned to the control or treatment groups. Tumor growth was evaluated by measurement twice a week of two perpendicular diameters of tumors with a caliper. Individual tumor volume, relative tumor volume (RTV), and tumor growth inhibition (TGI) were calculated according to standard methodology [51]. Moreover, to evaluate the overall response rate (ORR) to treatments observed in all treated models according to individual mouse variability, we decided to consider each mouse as one tumor-bearing entity. Hence, in all in vivo experiments, a relative tumor volume variation (RTVV) of each treated mouse was calculated from the following formula: [(RTVt/mRTVc)], where RTVt is the relative tumor volume of the treated mouse and mRTVc the median relative tumor volume of the corresponding control group at a time corresponding to the end of treatment. Then, for each treated mouse, we calculated [(RTVV)-1]. Finally, to clearly evaluate the impact of treatments on the tumor progression, we evaluated the probability of progression (doubling time and time for RTV × 4). 

In each in vivo experiment, frozen and formol-fixed tumor tissues were collected at the time of first ethical sacrifice in all treated groups, which depends on the tumor growth speed of each PDX. Between three to five tumors have been obtained from each group, according to the experimental design.

Statistical significance of observed differences between the individual RTVs corresponding to the treated mice and control groups was calculated using the two tailed Mann–Whitney test. Statistical significance of ORR between tested treatments was determined using a χ^2^ test. Predictive markers have been defined using a Mann–Whitney test. 

Studies have been performed in compliance with the recommendations of the French Ethical Committee and under the supervision of authorized investigators. The experimental protocol and animal housing followed institutional guidelines as put forth by the French Ethical Committee (Agreement number D-750602, France) and the ethics committee of the Institut Curie (Agreement number C75-05-18). 

### 4.4. Microarray Analysis—Transcriptome Arrays

mRNA gene expression of patient tumors and corresponding xenografts was defined using Human Exon 1.0 ST Array (Affymetrix, Santa Clara, CA, USA). RNA extraction, array hybridization, and analyses were performed as previously reported [29].

### 4.5. Reverse Phase Protein Array Study (RPPA)

RPPA methodology and analysis were performed as previously described [29]. Two different studies were performed: (i) A first one on 15 UM PDXs and their originating patient’s tumors identified in Némati et al. [17]; in this cohort, RPPA-based protein expressions were studied in the original patient tumors and corresponding xenografts at very early (P1), early (P4), and late (P9) in vivo passages in SCID mice. (ii) A second one on four to six tumors per group obtained at the end of four in vivo experiments evaluating the olaparib + DTIC combination (MP55, MP77, MM33, and MM52 PDXs). Differential abundance of the proteins between various tumor cohorts and groups was assessed using paired *t* tests assuming equal variance. The list of studied proteins and of used primary antibodies is presented in the Appendix A. 

### 4.6. Western Blot (WB) Analyses

Proteins were extracted as described previously [52]. Lysates were resolved on 4–12% TGX gels (Bio-Rad^®^, manufacturer, Marnes La Coquette, France). RNA extraction, array hybridization, and analyses were performed as previously reported [29].), transferred to nitrocellulose membranes (Bio-Rad^®^). Immunoblotting was performed with antibodies against the ubiquitous protein GAPDH and some other specific studied proteins, as shown in the Appendix A. After washes, membranes were incubated with the appropriate secondary horseradish peroxidase-conjugated affinity-purified goat anti-rabbit antibody (Jackson Immuno Research Laboratories, Inc., Interchim, San Diego, CA, USA). Quantification of protein expression was performed using Multi Gauge software and normalized to GAPDH expression. For each PDX model, the ratio of p-protein/protein in treated tumors was calculated. 

### 4.7. Immunohistochemical (IHC) Study

Immunohistochemical analyses were performed for the following pathways: Apoptosis and cell proliferation (caspase 3, cleaved-caspase 3, PARP, cleaved PARP, p-H2AX, and Ki67), the MAPK pathway (MEK1/2, p-MEK1/2, ERK, and p-ERK), the Pi3K pathway (AKT, p-AKT, S6, and p-S6), and the Hippo-related pathway (YAP and TAZ proteins). The list of all used antibodies is presented in the Appendix A. Paraffin-embedded tissue blocks and TMAs were cut with a microtome into fine slivers of 3 microns. Tissue sections were dewaxed and rehydrated through a series of xylene and ethanol washes before heat-induced epitope retrieval. Immunostaining was processed by using a Dako automated. The slides were incubated with primary antibody one hour at room temperature or overnight at 4 °C and then with secondary antibody coupled to horseradish peroxidase. For two antibodies (p-H2AX and cleaved-caspase3), the secondary antibody was biotinylated and the section incubated with ABC system. A DAB solution was applied for five minutes to reveal peroxidase. Slides were counterstained with hematoxylin before mounting with resin. A pathological score (0 to 3) was defined as % of positive cells (0 to 1) × intensity of immunostaining (0 to 3).

## 5. Conclusions

In conclusion, our work demonstrates that olaparib might be helpful in the treatment of metastatic UM patients when combined with dacarbazine. Such a combination should be tested after assessment of various predictive markers that we have also identified, and may be, if clinical tolerance is acceptable, associated with a Bcl-X_L_ inhibitor or a YAP inhibitor.

## Figures and Tables

**Figure 1 cancers-11-00751-f001:**
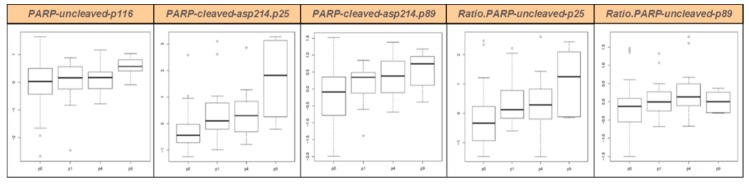
PARP protein expression in UM patients (P0) and their corresponding PDXs at various in vivo passages (P1, P4, and P9) as measured by RPPA. Boxplots represent the distribution of data of all PDX models together. Boxes contain 50% of values, upper brackets contain the 25% highest values, lower brackets contain the 25% lowest values. Black lines: median; dots: outliers. P0, patient’s tumor; P1, first in vivo passage; P4, fourth in vivo passage; P9, ninth in vivo passage.

**Figure 2 cancers-11-00751-f002:**
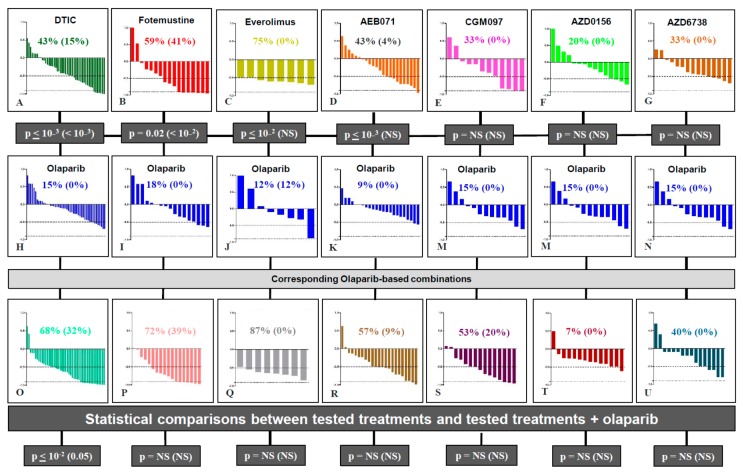
Waterfall plots of response to olaparib-based therapies. Upper *p* values correspond to statistical differences between olaparib compared to each other tested treatment; *p* values at the bottom correspond to statistical differences between each tested treatment compared to its combination with olaparib (χ^2^ test).

**Figure 3 cancers-11-00751-f003:**
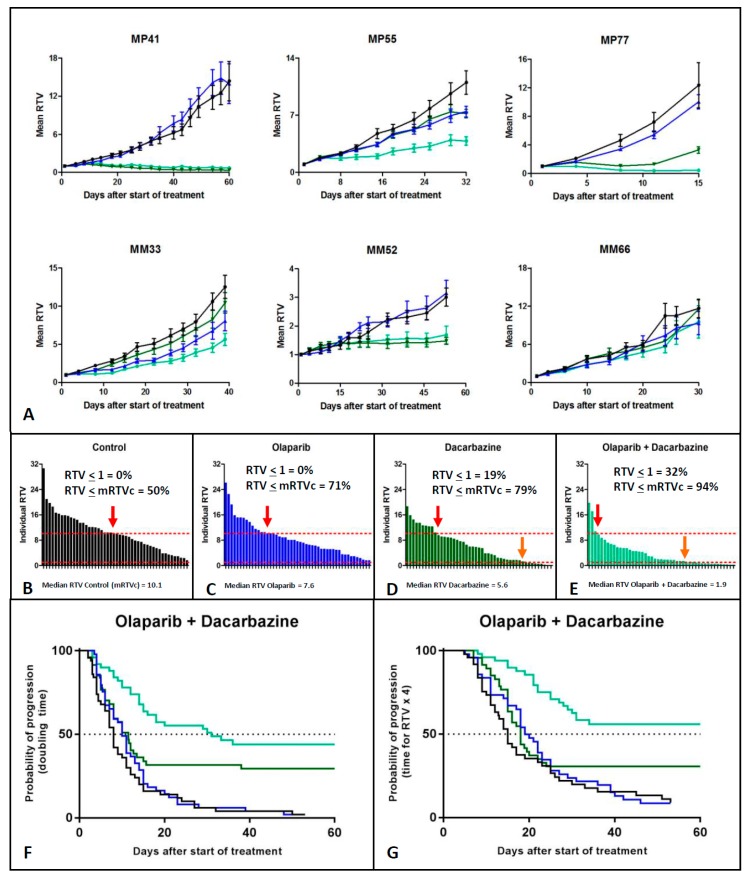
Combination of olaparib and dacarbazine. (**A**) Growth curves of all treated models. (**B–E**) ORR of the four experimental groups, i.e., control (black), olaparib (blue), dacarbazine (dark green), and olaparib + dacarbazine (light green), respectively. RTV: Relative tumor variation. mRTVc: median relative tumor volume of the corresponding control group (**F**–**G**). Probability of progression of the four experimental groups: doubling time (**F**) and quadrupling time (**G**).

**Figure 4 cancers-11-00751-f004:**
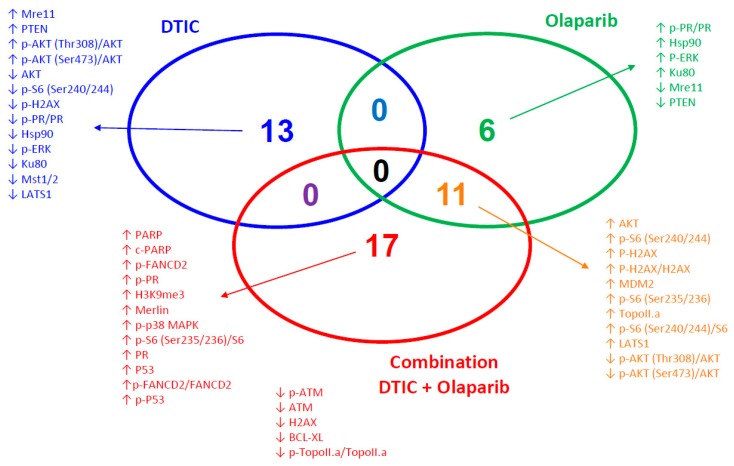
Venn diagram of the proteins significantly (*p* < 0.01) correlated to response to DTIC alone (blue) (MP55 + MP77 + MM52 versus MM33), olaparib alone (green) (MM33 versus MP55 + MP77 M MM52), or the DTIC+ olaparib combination (red) (MP55 + MP77 + MM33 versus MM52) (paired *t* tests assuming equal variance).

**Figure 5 cancers-11-00751-f005:**
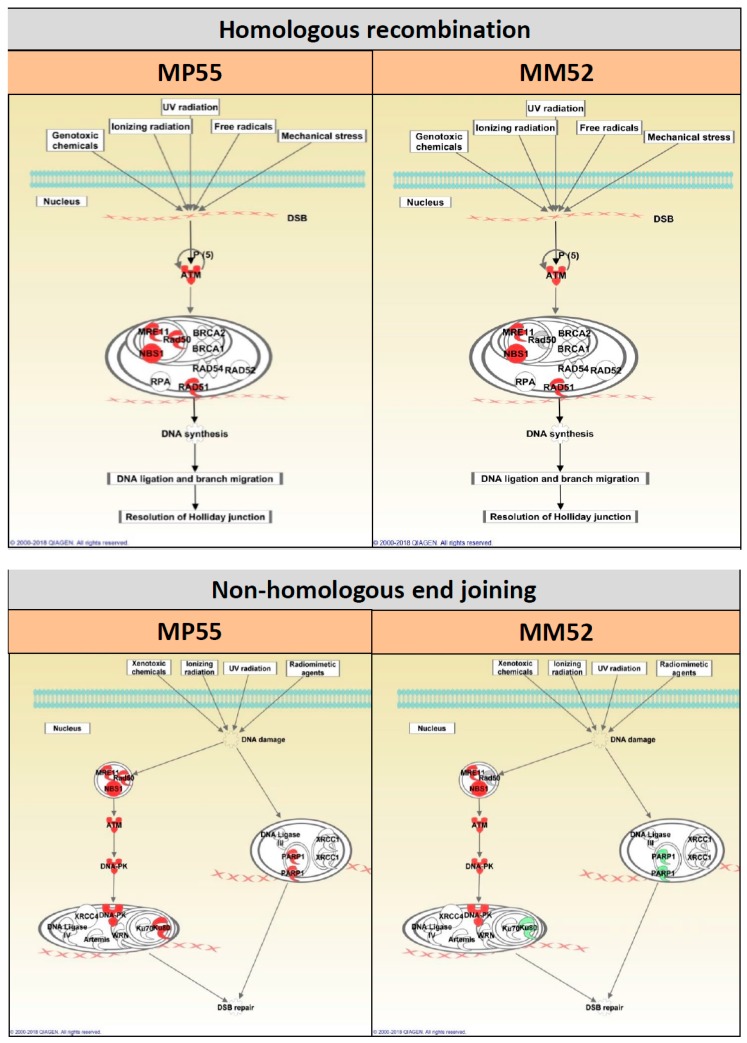
Schematic representation of HR- and NHEJ-related protein expression modifications in the two PDXs MP55 and MM52: Olaparib + DTIC versus DTIC. The intensity of color reflects fold change. Red color: higher (phospho-)protein expression in olaparib + DTIC than in DTIC alone. Green color: lower (phospho-)protein expression in olaparib + DTIC than in DTIC alone.

**Figure 6 cancers-11-00751-f006:**
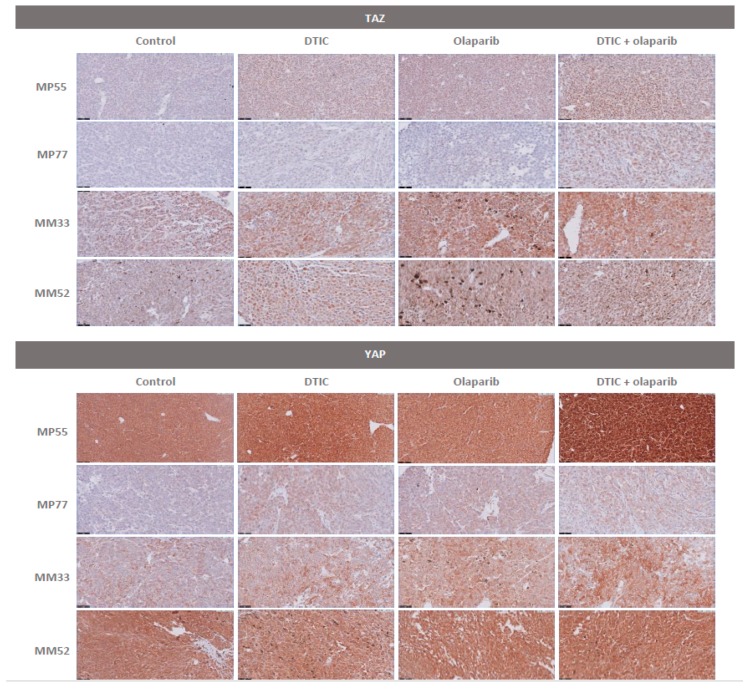
TAZ and YAP protein staining in the four treated PDXs by IHC.

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
