# Peer review of "PARP Inhibition Increases the Response to Chemotherapy in Uveal Melanoma"

_cancers, 2019, doi:10.3390/cancers11060751_

Reviewer 1 Report

The authors de Koning et al. present an interesting study on the expression of PARP genes and proteins in uveal melanoma (UM) patient samplesand their corresponding xenografts. Further, the authors undertook in vivo experiments evaluating the efficacy of olaparib alone or in combination with different chemotherapies and targeted drugs.

In summary, poly [ADP-ribose] polymerase (PARP) expression was observed in UM samples and the corresponding PDXs. The PARP inhibitor Olaparib was administered with and without previous chemotherapeutic agents employed previously in metastatic UM, including dacarbazine (DTIC), fotemustine, everolimus, the pan-PKC inhibitor AEB071, the MDM2 278 inhibitor CGM097 and the ATM inhibitor AZD0156. Essentially it was found that Olaparibsynergistically increased the effects ofDTIC in UM. Further the authors describe a strong and significant increase of both YAP and TAZ 258 protein expression and phosphorylation upon treatment with DTIC, and therefore they propose that YAP-inhibitors could be added to future ‘cocktails’ in the treatment of metastatic UM.

 The study is of value; however, it requires some modifications in the first instance. These include:

1.    In general, there could be a review of the written English for its fluency and to remove any small typographical errors. Further, the authors should check that they have defined an acronym before using it throughout the text – e.g. PARP. Indeed, this is not defined anywhere in text (!) despite PARP-inhibitors being the main focus of the paper. Similarly, the abbreviation DTIC should be defined on page 2, soon after the first mention of Dacarbazine, and then DTIC should be used consistently through the manuscript. Finally, Yes-associated protein (YAP) could also be defined at first mention in the text.

2.    In the Introduction, the authors could better outline the role of PARP proteins in cellular processes (in particular in DNA repair – in both single and double-strand breaks), and therefore point out the logic to the reader, as to why the PARP-inhibitors were used at all. Instead, the general reader at present is left to work out what PARP is, what these proteins do, etc. Similarly, the proposed importance of the Hippo-YAP/TAZ signalling pathway in UM should be explained better in the Introduction. Consider adding in further references, at the appropriate places in the text (e.g PMID: 31000600; PMID: 30773340).

3.    Figures: the resolution of the quite detailed and complicated figures varies, with it being quite pixelated in some – e.g. Figure 1 and 6 – making it quite difficult to interpret, even when using the zoom function on the PDF. In Figure 1, it appears that the error bars are very wide (PARP-uncleaved p-116, in the first panel), and it is unclear what the ‘‡’ is meant to represent in the second panel. In Figure 6, it cannot be determined where the immunostaining of both YAP and TAZ is located – what positive control (other than a UM cell line) was used to determine whether Ab used was appropriately staining the cell nucleus or cytoplasm? To be honest, at this resolution, I’m not sure what Fig 6 is meant to be demonstrating. Further, in some of the Figure legends (e.g. Figure 2 and some of the Supplementary figures/tables), the name(s) of the statistical test is missing. 

4.    How was the dosage of the various agents administered, alone and in combination, determined in the in vivoexperiments (S Table 11)? How did the authors determine whether the levels of the various drugs were sufficient? What side effects (if any) were seen? Did the authors consider other PARP inhibitors, e.g. ones that have been used in BRCA wild type tumors – e.g. Veliparib? 

5.    In the Discussion, the authors write about the association of the synergistic effect of Olaparib and DTICand a BAP-mutant cell line (MP55; Table S10), but suggest that the effect of Olaparib may be independent of BAP1. The authors cite the results of nuclear BAP1 negativity/positivity in mesothelioma and the proposals of epigenetic modification of the BAP1gene. There have been similar reports in UM, which the authors appear to have overlooked (e.g PMID: 29416875; PMID: 24633195; PMID: 27916271). Did the pattern of the nuclear BAP1 staining of the various UM cell lines and PDXs (BAP1 wt) change under therapy?

6. The authors suggest that YAP inhibitors (e.g. verteporfin) or a Bcl-XL inhibitor could be added to a regimen of PARP-Inhibitors and DTIC in metastatic UM: how do the authors envisage the time frame of the addition of these agents within the treatment regimen. This would obviously have to be tested within a clinical trial: have the authors considered the trial design?

Author Response

Reviewer 1:

Point 1: In general, there could be a review of the written English for its fluency and to remove any small typographical errors. Further, the authors should check that they have defined an acronym before using it throughout the text – e.g. PARP. Indeed, this is not defined anywhere in text (!) despite PARP-inhibitors being the main focus of the paper. Similarly, the abbreviation DTIC should be defined on page 2, soon after the first mention of Dacarbazine, and then DTIC should be used consistently through the manuscript. Finally, Yes-associated protein (YAP) could also be defined at first mention in the text.

Response of the authors:

-          English has been revised.

-          As requested, PARP, DTIC, and YAP abbreviations have been defined at first mention in the text of the manuscript (lines 51-52, 46, and 211-212 of the revised version of the article, respectively).

Point 2: In the Introduction, the authors could better outline the role of PARP proteins in cellular processes (in particular in DNA repair – in both single and double-strand breaks), and therefore point out the logic to the reader, as to why the PARP-inhibitors were used at all. Instead, the general reader at present is left to work out what PARP is, what these proteins do, etc. Similarly, the proposed importance of the Hippo-YAP/TAZ signaling pathway in UM should be explained better in the Introduction. Consider adding in further references, at the appropriate places in the text (e.g PMID: 31000600; PMID: 30773340).

Response of the authors: We thank the reviewer for this suggestion and improved the introduction with more background on PARP and on the Hippo pathway, which indeed strengthens the rationale of our manuscript (lines 73-79).

Point 3.1: Figures: the resolution of the quite detailed and complicated figures varies, with it being quite pixelated in some – e.g. Figure 1 and 6 – making it quite difficult to interpret, even when using the zoom function on the PDF.

Response of the authors: We believe that the low resolution is due to integration of the figures into the manuscript in a single pdf file. We are confident that the resolution will be satisfactory in the original images.

Point 3.2: In Figure 1, it appears that the error bars are very wide (PARP-uncleaved p-116, in the first panel), and it is unclear what the ‘‡’ is meant to represent in the second panel.

Response of the authors: Please note that these are not error bars, but boxplots. They thus do not represent the reproducibility of a repeated observation, but the distribution of data. 50% of the data points are within the box, the upper bracket represents 25% of the data and the lower bracket represents the 25% lowest data points. We agree with the reviewer that the biological variability of protein expression is large notably in patient samples (P0), which are more heterogeneous than xenografts. To the best of our knowledge, Figure 1 does not contain the “‡” symbol. We therefore cannot reply to this question.

Point 3.3:

In Figure 6, it cannot be determined where the immunostaining of both YAP and TAZ is located – what positive control (other than a UM cell line) was used to determine whether Ab used was appropriately staining the cell nucleus or cytoplasm? To be honest, at this resolution, I’m not sure what Fig 6 is meant to be demonstrating. Further, in some of the Figure legends (e.g. Figure 2 and some of the Supplementary figures/tables), the name(s) of the statistical test is missing.

Response of the authors:

Regarding YAP and TAZ IHC, the supplementary Figure S26 already shows nuclear and cytoplasm immunostaining. A sentence has been added in the article to clarify this point (lines 213-214). No positive control has been used, and negative control was validated by the absence of immunostaining of stromal mouse cells. Finally, as requested, statistical tests have been indicated in the Figure legends 2 and 4 (lines 150 and 181).

Point 4: How was the dosage of the various agents administered, alone and in combination, determined in the in vivo experiments (S Table 11)? How did the authors determine whether the levels of the various drugs were sufficient? What side effects (if any) were seen? Did the authors consider other PARP inhibitors, e.g. ones that have been used in BRCA wild type tumors – e.g. Veliparib?

Response of the authors:

The dosage of the various agents administered was defined according to published in vivo studies and tolerability of new tested combinations in mice. Hence, we did not observe any side effect in mice. Moreover, no other PARP inhibitor was considered in our study.

Point 5: In the Discussion, the authors write about the association of the synergistic effect of Olaparib and DTIC and a BAP-mutant cell line (MP55; Table S10), but suggest that the effect of Olaparib may be independent of BAP1. The authors cite the results of nuclear BAP1 negativity/positivity in mesothelioma and the proposals of epigenetic modification of the BAP1gene. There have been similar reports in UM, which the authors appear to have overlooked (e.g PMID: 29416875; PMID: 24633195; PMID: 27916271). Did the pattern of the nuclear BAP1 staining of the various UM cell lines and PDXs (BAP1 wt) change under therapy?

Response of the authors: Mesothelioma is cited here because it was found to be sensitive to PARP inhibition independently of BAP1 status, which might reflect what we see in our manuscript (although numbers are too low to formally conclude). The references indicated by the reviewer show that BAP1 protein expression can be lost without any mutation or copy number alteration, reinforcing the idea of potential epigenetic mechanisms. We now added these references into our manuscript (line 249).

Point 6: The authors suggest that YAP inhibitors (e.g. verteporfin) or a Bcl-XL inhibitor could be added to a regimen of PARP-Inhibitors and DTIC in metastatic UM: how do the authors envisage the time frame of the addition of these agents within the treatment regimen. This would obviously have to be tested within a clinical trial: have the authors considered the trial design?

Response of the authors:

Indeed, our results have highlighted two possible new therapeutic combinations, i.e. a PARP inhibitor + Bcl-XL or YAP/TAZ inhibitor. Such hypotheses might be confirmed in new in vivo experiments and, after confirmation, evaluated in metastatic UM patients. Both preclinical and clinical studies remain to be performed.

Reviewer 2 Report

This manuscript from de Koning et al describes the effect of PARP inhibition in the treatment of uveal melanoma. The manuscript altogether seems to be a useful and timely contribution and provides new insight into the pathogenesis of this malignancy.

The authors evaluated the expression of PARP proteins and the therapeutic effect of the PARP inhibitor olaparib, alone or in combination in UM Patient-Derived Xenogratfs (PDXs). The authors concluded based on in vivo experiments that olaparib in combination with dacarbazine was significantly efficient in the treatment of UM. Olaparib significantly increased the effect of the alkylating agent dacarbazine, however, such an effect could not be observed with other alkylating agents, e.g. fotemustine, everolimus, etc. Furthermore, they found various predictive biomarkers having an additive effect in response to olaparib and dacarbazine treatment.

This manuscript provides further support for a novel therapeutic approach of UM and opens new avenues in drug development and therapeutic strategies of the disease.

The experiments are described in sufficient detail.

However, some minor changes are needed and also some additional data would be required before accepting the manuscript. in its final version.

The manuscript needs some potential explanation about the signaling pathways and some background information why the combinational therapy (olaparib + dacarbazine) seems to be successful why others can not provide such good effects.

The authors also need to cite few additional references some of them published very recently:

1. Sato, T.; Han, F.; Yamamoto, A. The biology and management of uveal melanoma. Curr. Oncol. Rep. 200810, 431–438. 

2. Abildgaard, S.K.; Vorum, H. Proteomics of uveal melanoma: A mini review. J. Oncol. 20132013, 820953. 

3. Nichols, E.E.; Richmond, A.; Daniels, A.B. Tumor characteristics, genetics, management, and the risk of metastasis in uveal melanoma. Semin. Ophthalmol. 201631, 304–309.

4. Oláh G, Dobos N, Vámosi G, Szabó Z, Sipos É, Fodor K, Harda K, Schally AV, Halmos G. Experimental therapy of doxorubicin resistant human uveal melanoma with targeted cytotoxic luteinizing hormone-releasing hormone analog (AN-152). Eur J Pharm Sci. 2018 Oct 15;123:371-376. doi: 10.1016/j.ejps.2018.08.002. Epub 2018 Aug 2.

5. Harda K, Szabo Z, Szabo E, Olah G, Fodor K, Szasz C, Mehes G, Schally AV, Halmos G. Somatostatin Receptors as Molecular Targets in Human Uveal Melanoma. Molecules. 2018 Jun 26;23(7). pii: E1535. doi: 10.3390/molecules23071535.

6. A. Sharma, M.M. Stei, H. Fröhlich, F.G. Holz, K.U. Loeffler:  Genetic and epigenetic insights into uveal melanoma. Clinical Genetics 2018 Oct;94(3-4):398. doi: 10.1111/cge.13407

These papers are also focusing on other subpopulations of UM to identify novel molecular targets or potential therapeutic options. The authors should mention them in lines 49-50 and add these references (together with References 2 and 3).

In conclusion the manuscript is undoubtedly interesting and well written and the conceptual advance is pretty high thus it is a welcome addition to the literature.

Author Response

Reviewer 2:

Point 1: The manuscript needs some potential explanation about the signaling pathways and some background information why the combinational therapy (olaparib + dacarbazine) seems to be successful why others cannot provide such good effects.

Response of the authors: We now added more background on PARP and Hippo pathways in the introduction (lines 73-79 and 52, respectively). Due to DNA lesions induced by the alkylating agent dacarbazine, it was not surprising that olaparib, which impacts DNA repair, increases antitumor activity of the chemotherapeutic agent.

Point 2: The authors also need to cite few additional references some of them published very recently.

Response of the authors:

We strongly thank the reviewer for suggesting various references that may enhance the scientific quality of the article. However, some of these articles are quite far from our study which is not a review on uveal melanoma. To maintain a focus on our work, we therefore propose to not add the two articles evaluating somatostatin (Harda et al) and LHRH receptor (Olah et al) as potential therapeutic approaches. In contrast, the four other articles proposed by the reviewer have been included in the revised version of the article, at the appropriate places in the text (line 49).

Point 3: These papers are also focusing on other subpopulations of UM to identify novel molecular targets or potential therapeutic options. The authors should mention them in lines 49-50 and add these references (together with References 2 and 3).

Response of the authors:

As requested, some references have been added (line 49)

Reviewer 3 Report

The manuscript submitted by Koning el reports on their work utilizing a panel of 12 patient tumors and 32 uveal melanoma PDX models to assess the expression of PARP proteins and the antitumor activity of olaparib alone and in combination with a number of different agents.  They demonstrate stable PARP expression between patient tumors and the corresponding PDX, the combinatorial activity of olaparib and DTIC, and identified potential predictive biomarkers of response to this regimen.  The authors present a tremendous number of interesting experiments within this manuscript, although, in part due to this volume of data, the manuscript at times is difficult to follow.  In general, improved annotation of the figures would assist with clarity and readability.  My specific comments are as follows:

1. Figure S1. Potential biological and clinical implications of the variable PARP family gene expression should be included within the manuscript. For instance, what relevance, if any, is there regarding the low expression of PARP15 and PARP16?

2. Figure 1, bottom panels. This may be more appropriate as a supplemental figure.

3. Section 2.2. Rational for the selection of agents to test in combination with olaparib should be provided. Furthermore, an explanation for why a different number of models and why a different number of mice per model were utilized across different experiments should be provided in this section.

4. Figure 2. The text within the “black” rows (including the p value rows) are impossible to read in part.  It is not possible to decipher for which regimen each waterfall plot applies.  Labeling each waterfall plot with the RR might be helpful.  Within the text, please clarify how response is defined?  Is this 50% reduction in tumor volume?  Or greatest diameter?  Finally, the 75% response rate for everolimus is interesting… has there been any followup on this finding?

5. Section 2.3. Provide an explanation as to why the MP21 and MP66 models were excluded from the RPPA analyses.

6. Was formal synergy analyses performed to assess the additive/synergistic effects of olaparib and DTIC?

7. Section 2.3. Further details regarding the response rate of DTIC alone, olaparib alone, and the combination in each of the 4 PDX models utilized should be provided to help assess the robustness of the RPPA analyses.

8. Figure 4. Although the clinical response rate of uveal melanoma to DTIC is low, it is not 0%. These data suggest a potential signature for responsiveness to DTIC which should be included within the discussion.

9. BAP1 expression by IHC should be performed in all PDX models to allow correlation with treatment response findings (as opposed to BAP1 mutations alone).

Author Response

Reviewer 3:

Point 1: Figure S1. Potential biological and clinical implications of the variable PARP family gene expression should be included within the manuscript. For instance, what relevance, if any, is there regarding the low expression of PARP15 and PARP16?

Response of the authors: Although there is no consensus yet, it has been suggested that the expression or activity of PARP proteins could be predictive for the response to PARP inhibition (reviewed in Bitler et al, Gynecologic Oncology, 2017). Our main objective here was therefore to validate that our PDX models reliably reflect patient samples in terms of PARP expression levels and activation status. We show that this is the case, at least for the PARP1–4 proteins which are the ones targeted by Olaparib (Wahlberg et al. Nature Biotechnology 2012). The other PARP proteins are currently not well studied (Bai, Molecular Cell 2015) and it is not clear whether their expression levels influence on the efficiency of PARP inhibition in vivo. We can thus only hypothesize about their potential biological and clinical implications. We now added this into the discussion (line 225).

Point 2: Figure 1, bottom panels. This may be more appropriate as a supplemental figure.

Response of the authors:

As requested, the bottom panels of the Figure 1 have been moved as a supplemental figure (new Figure 1 and new Figure S1) (Lines 100.111-112).

Point 3: Section 2.2. Rational for the selection of agents to test in combination with olaparib should be provided. Furthermore, an explanation for why a different number of models and why a different number of mice per model were utilized across different experiments should be provided in this section.

Response of the authors:

As requested, rationale for the selection of agents to test in combination with olaparib has been added in the section 2.2 (lines 117-123). Moreover, due to the fact that this study was conducted during a relative long period, about two years of in vivo experiments, the number of models used for each combination of treatments and the number of mice per model was not homogeneous along the study. This sentence has been added in the M&M section 4.3 (lines 304-306).

Point 4: Figure 2. The text within the “black” rows (including the p value rows) are impossible to read in part.  It is not possible to decipher for which regimen each waterfall plot applies.  Labeling each waterfall plot with the RR might be helpful.  Within the text, please clarify how response is defined?  Is this 50% reduction in tumor volume?  Or greatest diameter?  Finally, the 75% response rate for everolimus is interesting… has there been any followup on this finding?

Response of the authors:

As requested, the Figure 2 has been modified to better indicate regimen. The modality to define response has already been mentioned in the article (M&M section 4.3). As it was also defined, such a definition considered tumor volume and not tumor diameter. Considering the high response to everolimus, we had already published this observation in our UM PDXs (reference 20) and, to our knowledge, no clinical data have been reported in metastatic UM patients, yet.

Point 5: Section 2.3. Provide an explanation as to why the MP21 and MP66 models were excluded from the RPPA analyses.

Response of the authors:

The MP21 and MP66 models have been treated later in time, after the onset of the RPPA study. Hence, they could not be included in RPPA analyses.

Point 6: Was formal synergy analyses performed to assess the additive/synergistic effects of olaparib and DTIC?

Response of the authors:

No formal in vitro analyses have been performed. Due to the fact that our UM PDXs very well reproduce their patient’s originating tumors, as previously published (Némati et al, CCR 2010; Laurent et al, Mol Oncol 2013), we have considered that in vivo experiments –performed in UM PDXs – were superior to in vitro analyses. Moreover, in a previous report (Carita et al, Oncotarget 2016), we have combined both in vitro and in vivo experiments to evaluate combinations of treatments in UM PDXs and in this article we showed that in vitro synergistic effect could be not demonstrated despite high in vivo efficiency of the tested combination (PKC inhibitor AEB071 + MDM2 inhibitor CGM097, in comparison to each tested monotherapy).

Point 7: Section 2.3. Further details regarding the response rate of DTIC alone, olaparib alone, and the combination in each of the 4 PDX models utilized should be provided to help assess the robustness of the RPPA analyses.

Response of the authors: The response rate of each of the 4 PDX models is already detailed in supplementary table S4.

Point 8: Figure 4. Although the clinical response rate of uveal melanoma to DTIC is low, it is not 0%. These data suggest a potential signature for responsiveness to DTIC which should be included within the discussion.

Response of the authors: We agree with the reviewer that the proteins in blue in Figure 4 might constitute predictive biomarkers for response to DTIC alone. We now added a phrase on this in the discussion (lines 251-253).

Point 9: BAP1 expression by IHC should be performed in all PDX models to allow correlation with treatment response findings (as opposed to BAP1 mutations alone).

Response of the authors: In a previous publication on our UM PDXs (Laurent et al), which is already referenced in the manuscript, we have reported BAP1 expression by IHC in 15 models (Laurent et al, Table S6), among which all six PDXs treated by the combination of olaparib + DTIC in the current manuscript. Hence, we have clearly established a high correlation between the presence of a BAP1 gene mutation and the absence of BAP1 protein expression (p < 0.01) (Laurent et al, Figure 3). In brief, BAP1 is WT with intense nuclear staining in MP41, MP77, MM33 and MM66 models. In the two BAP1 mutated models, MP55 and MM52, BAP1 staining is absent and faint cytoplasmic, respectively. When available, data of BAP1 IHC expression have been added in the Table S10. There is thus a good correlation between BAP1 mutation status and protein expression in our models and no indication of epigenetic regulation. We now added this in the discussion (lines 245-246).

Reviewer 4 Report

Manuscript ID: cancers-473427

The authors present an in vitro patient derived xenograft model to test for the possible efficacy of addition of a PARP inhibitor to alkylating chemotherapy for metastatic uveal melanoma. Based on the supposed effects of BAP1 on DNA repair such an additive effect could especially be expected in those tumors that show loss of BAP1 expression. First the authors describe high expression of PARP1, -4, -6, -10, -12 and -14 in the primary tumors that have been used for the xenograft models. The expression of these genes is not significantly altered in the in vitro passages. Next, 6 in vivo xenograft models have been treated by two different alkylating agents, alone or in combination with the PARP inhibitor Olaparib. In three models, two of a primary tumor and on of a metastasis, an additive effect was observed for the combination of Olaparib and dacarbazine, but not for Olaparib and fotemustine. Only one of these three tumors had loss of BAP1.  The three experiments that did show an additive effect were then compared to those that did not using proteomics with the aim to identify factors that would predict such an additive response. The effects of alkylating chemotherapy in combination with PARP inhibition could not be ascribed to apoptosis related pathways or MAPK or Pi3K pathways. The authors did observe an increase in phosphorylated YAP but this was not related to combination therapy. Overall this is an extensive study that may indicate some interesting new treatment options for patients suffering metastatic uveal melanoma. However, the study is complicated by superfluous data and lacks focus on the primary incentive which is to evaluate the effect of PARP inhibition in addition to alkylating chemotherapy in tumors that show loss of BAP1 expression. The lack of effect of the combination of Olaparib and fotemustine was not further investigated. Few comments can be made.

Major comments

It is unclear to the reviewer why the combination of Olaparib and everolimus, AEB071, CGM097, AZD0156 and AZD6738 was tested in the xenograft model. An additive effect is not expected for these treatments and they have precluded a clear conclusion on the differences between the alkylating agents dacarbazine and fotemustine for which the animals might have been put to better use.

This study was started with the hypothesis that BAP1 loss might be of importance for DNA repair in uveal melanoma, however the tumors that have been used for the study do not seem to have been selected based on BAP1 status. Can the authors provide a calculation of the expected number of cases to treat needed to show a difference and can they explain why this could not be met.

Minor comments

Abstract, page 1: the abstract needs extensive rewriting. The formulations are vague and do not clearly describe the studies performed. Terminology such as “various analyses”, “various predictive markers” should not be used.

Introduction page 2, lines 80-86: please remove this is for the discussion.

Conclusions, page 11, line 358: the combination of the words “clearly” and “might” in one statement is a contradiction in terms. Please rephrase.

Author Response

Reviewer 4:

Point 1: It is unclear to the reviewer why the combination of Olaparib and everolimus, AEB071, CGM097, AZD0156 and AZD6738 was tested in the xenograft model. An additive effect is not expected for these treatments and they have precluded a clear conclusion on the differences between the alkylating agents dacarbazine and fotemustine for which the animals might have been put to better use.

Response of the authors: We agree that it was not surprising that olaparib was efficient in combination with the alkylating agent DTIC, however we could not exclude additive effects when combined with targeted therapies such as everolimus, AEB071, CGM097, AZD0156, and AZD6738. Indeed, everolimus, AEB071, and CGM097 are treatments that were previously successful in UM PDXs (Amirouchene-Angelozzi N et al, 2016; Carita et al, 2016) and for which clinical trials are ongoing; AZD0156 and AZD6738 are inhibitors of DNA repair enzymes that  increase olaparib efficiency (Pike et al, 2018; Kim et al, 2017). Those experiments therefore constituted an interesting part of our in vivo study.

Point 2: This study was started with the hypothesis that BAP1 loss might be of importance for DNA repair in uveal melanoma, however the tumors that have been used for the study do not seem to have been selected based on BAP1 status. Can the authors provide a calculation of the expected number of cases to treat needed to show a difference and can they explain why this could not be met.

Response of the authors: A potential role for BAP1 in DNA repair was indeed one of the hypotheses that led us to test Olaparib. However, as mentioned in the discussion, PARP inhibition can also potentiate the response to chemotherapeutic agents in the absence of DNA repair deficiencies. Typically, in malignant pleural mesothelioma, which resemble uveal melanoma in some aspects, Olaparib seems to affect proliferation independently of BAP1 status (Srinivasan et al 2017). We therefore did not want to concentrate solely on BAP1 mutant models.

In case we would like to formally determine whether the additive effect of DTIC + Olaparib is dependent or not on BAP1 status, we would probably need to establish a contingency table followed by a Chi2-test, for which the general recommendation is to test at least 20 models, among which at least 5 BAP1 WT, 5 BAP1 mutants, 5 models showing an additive effect and 5 models without an additive effect. We currently do not have the required PDX models to test this hypothesis.

Point 3: Abstract, page 1: the abstract needs extensive rewriting. The formulations are vague and do not clearly describe the studies performed. Terminology such as “various analyses”, “various predictive markers” should not be used.

Response of the authors: The abstract was extensively rewritten and vague formulations were removed and replaced by precise information.

Point 4: Introduction page 2, lines 80-86: please remove this is for the discussion.

Response of the authors: We now removed results from the introduction and only introduce our approach here (lines 84-90).  

Point 5: Conclusions, page 11, line 358: the combination of the words “clearly” and “might” in one statement is a contradiction in terms. Please rephrase.

Response of the authors: This sentence was rephrased (line 375).

Round  2

Reviewer 1 Report

The authors have improved their manuscript in line with the reviewers' suggestions.

Reviewer 4 Report

the reviewers comments have been addressed adequately